



# Architectural Insights and Training Methodology Optimization of Pangu-Weather

Deifilia To[1], Julian Quinting[2], Gholam Ali Hoshyaripour[2], Markus Götz[1,3], Achim Streit[1], and Charlotte Debus[1]

[1]Karlsruhe Institute of Technology (KIT) - Scientific Computing Center (SCC)
[2]Karlsruhe Institute of Technology (KIT) - Institute of Meteorology and Climate Research
[3]Helmholtz AI

**Correspondence:** Deifilia To (deifilia.to@kit.edu)

**Abstract.** Data-driven medium-range weather forecasts have recently outperformed classical numerical weather prediction models, with Pangu-Weather (PGW) being the first breakthrough model to achieve this. The Transformer-based PGW introduced novel architectural components including the three-dimensional attention mechanism (3D-Transformer) in the Transformer blocks and an Earth-specific positional bias term which accounts for weather states being related to the absolute position on Earth. However, the effectiveness of different architectural components is not yet well understood. Here, we reproduce the 24-hour forecast model of PGW based on subsampled 6-hourly data. We then present an ablation study of PGW to better understand the sensitivity to the model architecture and training procedure. We find that using a two-dimensional attention mechanism (2D-Transformer) yields a model that is more robust to training, converges faster, and produces better forecasts than with the 3D-Transformer. The 2D-Transformer reduces the overall computational requirements by 20–30%. Further, the Earth-specific positional bias term can be replaced with a relative bias, reducing the model size by nearly 40%. A sensitivity study comparing the convergence of the PGW model and the 2D-Transformer model shows large batch effects: however, the 2D-Transformer model is more robust to such effects. Lastly, we propose a new training procedure that increases the speed of convergence for the 2D-Transformer model model by 30% without any further hyperparameter tuning.

## 1 Introduction

Data-driven medium-range weather forecasting models have experienced a rapid progress over the last years. Pangu-Weather (PGW) was the first model to surpass the performance of European Center for Medium-Range Weather Forecast (ECMWF)'s numerical weather prediction (NWP) method, the Integrated Forecasting System (IFS). Since then, several other models have been released, including GraphCast (Lam et al., 2023), FengWu (Chen et al., 2023a) and FuXi (Chen et al., 2023b). Increasing interest in the field has also led to the development of foundation models such as ClimaX (Nguyen et al., 2023), AtmoRep (Lessig et al., 2023), and Aurora (Bodnar et al., 2024). Many of these models introduce a number of unique architectural components. However, due to the enormous costs of training them, the publications often lack thorough ablation studies, making it difficult to determine which of the architectural components lead to the success of the models. This is particularly noteworthy in PGW, which contains 64 million parameters and requires an estimated 73000 GPU-hours on NVIDIA V100s



per lead time to train. The authors also admit that the models have not arrived at full convergence, and the large size of the model prohibits thorough hyperparameter optimization. In this work, we focus on PGW, as it constitutes a major breakthrough model for the field, but neither the architecture nor training procedure are well understood.

The authors of PGW published their trained model weights and pseudocode (Bi et al., 2023b), which cannot be run. The replicated implementations available on GitHub are not designed in a modularized manner such that substitution of different architectures is possible. To the best of the author's knowledge, no ablation study of PGW has been published to date. Willard et al. (2024) conducted an ablation study of Transformer-based models for end-to-end weather prediction based on a SwinV2-Transformer (Liu et al., 2021) implementation and investigated the effects of model size, constant channel weighting, and multi-step fine tuning.

In this work, we replicate the 24 h model of PGW trained on a 6-hourly subset of the data. Then, we perform an ablation study to determine which of the architectural components contribute most to the performance of the model. We notice that a 2D-Transformer model converges more quickly and results in lower RMSE values than the proposed 3D-Transformer model. Through our ablation study, we show that global batch size is a hyperparameter that strongly affects how quickly the model converges. We also notice that all variants of the model appear to have two "phases" of training: one that quickly stagnates until the model learns a good representation of the input, and then another phase in which the losses rapidly drop, even without adjusting the learning rate or optimizer. We argue that finding a training procedure that shortens the length of time in "phase 1" is crucial, as it can drastically reduce the total computational power and time required to train these models, allowing more resources to be invested in fine-tuning the model; another benefit is increasing the accessibility of training these models to researchers who only have access to minimal computational resources. With this in mind, we propose a new training procedure that increases the speed of convergence for the 2D-Transformer model by 30% without adjusting any other hyperparameters, increasing the possible performance of the model given a fixed compute budget.

## 2 Method

### 2.1 Data

The ECMWF Reanalysis v5 (Hersbach et al., 2020) dataset obtained from WeatherBench2 (Rasp et al., 2023) was used. As with PGW and common in other published models, the variables used were U-velocity (U), V-velocity (U), temperature (T), specific humidity (Q) and geopotential (Z) on 13 pressure levels (1000 hPa, 925 hPa, 850 hPa, 700 hPa, 600 hPa, 500 hPa, 400 hPa, 300 hPa, 250 hPa, 200 hPa, 150 hPa, 100 hPa, 50 hPa). The surface variables were mean sea level pressure (MSP), 10 m U-velocity (U10), 10 m V-velocity (V10), and 2 m temperature (T2M). Constant masks of soil type, topography and land masks were also included as input. All of the data was normalized by subsampling 6-hourly data and calculating the mean and standard deviation across all of the variables. Subsampling was required because of the high computational cost of training on the full dataset. Z-score normalization was applied. The Pangu-Weather model with a lead time of 24 h was trained on subsampled 6-hourly data until the model matched or performed better than IFS according to the RMSE at lead times of 3 and 5 days for all variables.





## 2.2 Compute infrastructure

We ran all experiments on a distributed-memory, parallel hybrid supercomputer. Each compute node is equipped with two 38-core Intel Xeon Platinum 8368 processors at 2.4 GHz base and 3.4 GHz maximum turbo frequency, 512 GB local memory, a local 960 GB NVMe SSD disk, two network adapters, and four NVIDIA A100-40 GPUs with 40 GB memory connected via NVLink. Inter-node communication uses a low-latency, non-blocking NVIDIA Mellanox InfiniBand 4X HDR interconnect with 200 GB s$^{-1}$ per port. All experiments used Python 3.9.16 with `CUDA`-enabled `PyTorch` 2.1.1. (Paszke et al., 2019).

## 2.3 Reproduction of Pangu-Weather and model description

The Pangu-Weather (PGW) code was reproduced according to the details provided in the paper (Bi et al., 2023a) and based on the pseudocode released by the authors (Bi et al., 2023b). Code is available at https://github.com/DeifiliaTo/PanguWeather (To, 2024). Our code was written in a modular manner such that different architectural details such as depth, hidden dimension, bias terms, and dimensionality of attention in the Transformer blocks can be altered.

PGW is an autoregressive model and uses a hierarchical temporal aggregation method to roll out the forecasts—four independent models, with lead times of $\Delta t = 1, 2, 6, 24$ hours, are trained. The authors of PGW introduced a three-dimensional-Transformer and was built on a Sliding Window architecture (SWIN) (Liu et al., 2021). The architecture of PGW consists of a patch embedding step, where the 3D field data are divided into 3D patches of shape (2, 4, 4). A 3D convolution is performed over the individual patches. Then, the patches are fed through a 3D-Transformer block, in which attention is computed over 3D windows (in the longitude, latitude, and vertical dimensions) over several layers. In contrast, the original SWIN architecture was implemented as a 2D-Transformer. After one transformer block, the patches are downsampled by a factor of (1, 2, 2), and passed through a linear layer, where the hidden dimension is increased by a factor of two. The states are then passed through two more 3D-Transformer blocks before being upsampled and recovering the patches through a transpose convolutional layer. PGW also introduces an Earth-specific position bias (ESB) term, which represents a learnable bias term added to the attention layers (Bi et al., 2023a). The "Earth-specific" terminology refers to the fact that the bias term is independently learned in the latitude and vertical dimensions. Due to the large number of layers, the ESB is a huge parameter, comprising 41 million parameters per lead time.

As validation case, the full-sized Pangu-Weather model with a lead time of 24 hours was traineduntil the model matched or performed better than IFS according to the RMSE at lead times of 3 and 5 days for all variables. Data from 1979–2017 was used as training, data from 2019 was used as validation, and data from 2020–2021 was testing.

## 2.4 Training procedure

The validation case was trained with a local batch size of two on 120 A100-40 NVIDIA GPUs. Unless otherwise specified, all ablated models were trained with a local batch size of six on 120 A100-40 NVIDIA GPUs. PyTorch's ReduceLROnPlateau scheduler was used with an initial learning rate of 0.0005, a patience of 7 and a factor of 0.5. The Adam optimizer was used with weight decay of $3 \cdot 10^{-6}$. Gradient checkpointing was used to increase the local minibatch size.





| Model | Number of parameters (millions) |
|---|---|
| Absolute bias (original) | 44.6 |
| Relative bias | 24.3 |
| Positional embedding | 24.3 |
| 2D-Attention | 57.2 |
| Three-depth | 108.9 |

**Table 1.** Overview of models included in ablation study. All models were based on PGW-Lite.

## 2.5 Ablation study

Due to computational constraints, an ablation study was performed based on the PGW-Lite architecture described by the authors rather than the full PGW model. The PGW-Lite retains all architectural components as the full-sized model, except for the first embedding step, in which a (2, 8, 8) patch size is used instead of a (2, 4, 4) patch size. This reduces the size of the model from 64 million to 40 million free parameters. As explained in Rajbhandari et al. (2019), the memory requirement of training is much greater than the size of the model, since intermediate model states are stored in the forward and backward passes.

Reducing the model size allows a larger local batch size to fit onto a single GPU, reducing the overall computational cost of training. For all ablated models, 6 h-subsampled data from 2008–2018 were used as training data, 2019 data was validation, and 2020–2021 was reserved for testing. The models were trained to produce 24 h forecasts. In the ablation study, five models were compared: These models and the number of parameters are shown in Table 1 and are described in more detail as follows:

In the PGW-Lite model, an absolute ESB term within the Transformer was implemented according to the description in Bi

et al. (2023a). The formulation of self attention in the PGW model is

$$\text{Attention}\left(\mathbf{Q}, \mathbf{K}, \mathbf{V}\right) = \text{SoftMax}\left(\mathbf{Q}\mathbf{K}^T/\sqrt{D} + \mathbf{B_{ESP}}\right)\mathbf{V} \tag{1}$$

where **Q, K, V** are the query, key, and value vectors, and D is the feature dimensionality of Q. $\mathbf{B_{ESP}}$ was introduced by Bi et al. (2023a) to represent a bias term that varies as a function of latitude and height but is invariant to longitude. Bi et al. (2023a) reasoned that atmospheric fields are variable in the latitude and vertical dimensions but remain consistent in

the longitudinal dimension. Even in the Lite model, this term contains 20 million parameters, comprising 50% of the overall terms in the model. In the relative bias model, the bias is invariant in all three dimensions, reducing the model size to 24.3 million parameters. In the positional encoding model, the patches are given a learnable positional encoding term implemented similarly to PGW-Lite, in that the patches in the latitude and vertical dimensions learn a positional embedding with a hidden dimension of $C = 192$, and the values stay the same along the longitude dimension. As this term is completely outside of the

attention blocks, it requires much fewer parameters to implement. The three-depth model increases the depth of the network to three, i.e., two up- and downsampling layers were implemented. To keep the model size manageable, only four layers within the deepest block were implemented. As with the original model, the downsampling was implemented in the latitude and longitude dimensions by patches of (2, 2) and increasing the hidden dimension by fourfold with a linear layer. In the 2D-Transformer



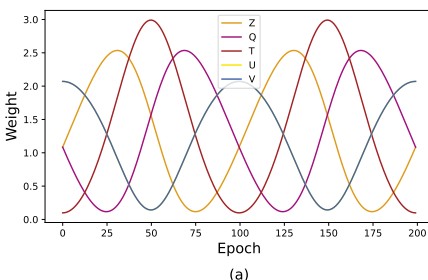
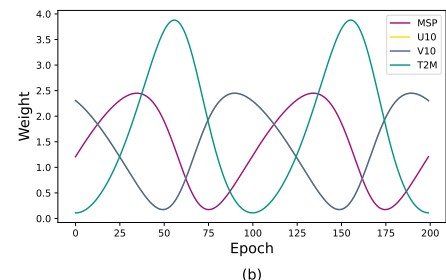

**Figure 1.** The weights for variable-specific cosine loss scheduling. The period is 100 epochs. The U- and V-velocity fields for both the pressure-level variables and surface variables are superimposed on one another, since they always receive the same values. The weights are normalized to sum up to the sum of the original weights presented in Bi et al. (2023a). (a) Pressure-level variables. (b) Surface variables.

model, the structure of PGW-Lite was retained, but the atmospheric variables of the different pressure levels were treated as

different variables (channels). The attention mechanism in the Transformer block was only applied over values in the longitude and latitude dimensions. To compensate for the reduction in parameters associated with a 2D-Transformer model, the hidden dimension was increased from $C = 192$ to $C = 288$, increasing the dimension of the hidden layer to 48 per attention head as opposed to the original 32. Note that the latent space in the 2D-Transformer now encodes 5 variables × 13 pressure levels instead of just 5 variables × 1 pressure level as in the 3D case.

**2.6 Variable-specific weighted cosine loss scheduling**

The loss function in PGW is defined as

$$\text{Loss} = \sum_i w_i \cdot \text{L1} \left( \text{prediction}, \text{target} \right) \tag{2}$$

where $i$ is the variable, $w$ is a weighting factor, L1 is the L1-loss function, prediction are the model outputs and target is the ground-truth forecast data. The values of $w_i$ are $3.00, 0.6, 1.5, 0.77$, and $0.54$ for Z, Q, T, U, V, and surface variable weights of

$1.5, 0.77, 0.66$, and $3.00$ for variables MSP, U, V, T, as specified by Bi et al. (2023a).

After determining that the 2D-Transformer model appears to converge the fastest and require the fewest computational resources, we re-train the 2D-Transformer model from scratch for 300 epochs with a smooth loss function that weights the variables differently over the epochs. The loss function is designed so that within 100 epochs, the weights for each variable cycles through one cosine schedule (Fig. 1). The peaks of the weights for each variable are scheduled equidistant in time.

At each epoch, the pressure variables are normalized to sum up to the original weights of $3.00 + 0.60 + 1.50 + 0.77 + 0.54$. The same is done for the surface variables. The validation loss is always calculated with the original weights to maintain consistency. Note that the two schedules for pressure level and surface variables are complementary: the weights for velocity fields and temperature at both levels are in phase.





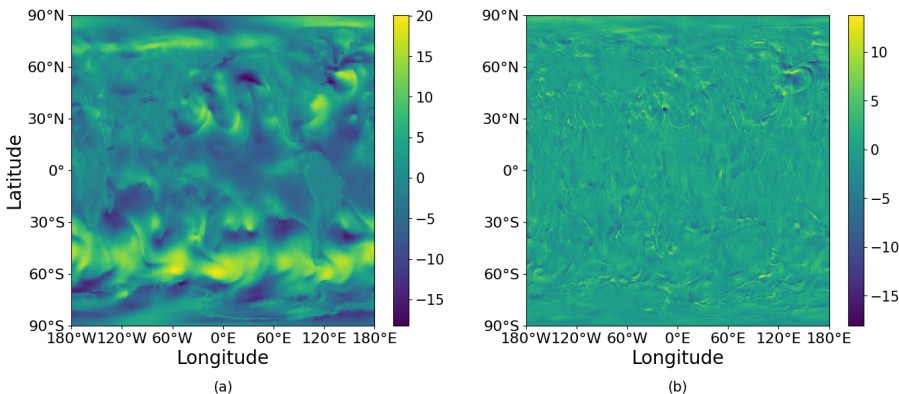

**Figure 2.** Representative sample of the U-velocity field at a pressure level of 1000 hPa for a 24 h forecast. The forecast was initialized 26 Jan 2020 on 00:00 UTC. (a): U-velocity field. Colorbar represents the U-velocity in units m/s; (b): Absolute error. Colorbar represents the absolute error between the prediction and the ground truth.

## 3 Results

### 3.1 Validation

A representative output of the model for a 24 h forecast as well as the absolute error is visualized in Figure 2.

Figure 3 shows the RMSE with increasing lead time of two surface variables, U-velocity at 10 m, and the temperature at 2 meters, as well as temperature at a pressure level of 850 hPa, tested over the years 2020 and 2021. We observe that our PGW model performs better than IFS for all three variables over the five-day forecast. There is still a notable difference between the performance of our model and the published PGW model. We attribute that to training the model on only a 6 h-subsample of the total data trained by the original authors, as well as slight differences in training procedure such as batch size. However, this result gives us enough confidence that complex weather patterns can be learned with our model, such that we proceed with the ablation study.

### 3.2 Ablation study

The training and validation loss of the different models are shown in Figure 4. All of the validation losses closely parallel the training losses, indicating that none of the models are overfitted. Despite drastic variations in the model architecture and in the size of the networks, all models exhibit similar loss curves. Furthermore, they reach similar final loss values upon model convergence.

As the loss function for the model is the L1 loss, we also wish to verify that meteorologically relevant metrics such as the RMSE and the anomaly correlation coefficient are improving for specific variables as the L1 loss goes down. Figure 4 shows the weighted RMSE values of the U-velocity at 10 m. Though all models learn with a similar structure, we note that the 2D-Transformer is able to reach some of the lowest RMSE values at a given epoch, i.e. compute budget.



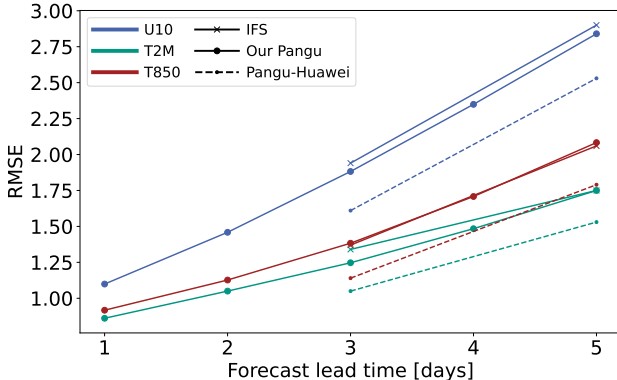

**Figure 3.** RMSE as a function of forecast lead time of key prognostic variables (T2M, U10, T850) in our model compared to IFS and Pangu-Weather on testing data from 2020–2021.

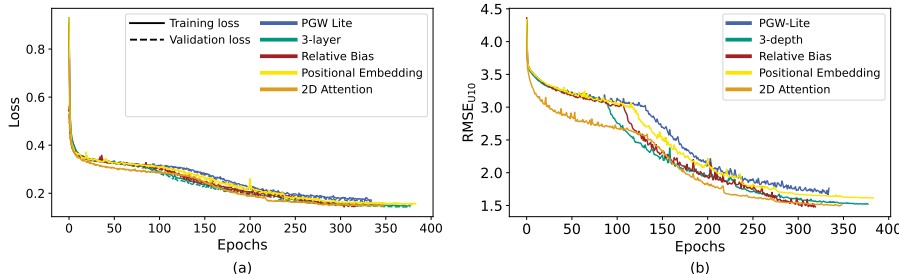

**Figure 4.** Training curves of ablation study for PGW-Lite, three-depth model, relative bias, positional embedding, and 2D-Attention models. (a): Training and validation loss values as a function of the epochs. (b): RMSE for the 10m-U-velocity as a function of the epochs, as evaluated for the validation samples.

The best validation loss as well as the epoch in which that is reached is shown in Table 2. All models reach a similar validation loss, with the value for the positional embedding being the worst at 0.152, and the PGW-Lite, relative bias, and three-depth tying for the best at 0.143. However, it is important to note that the 2D model converged 40–100 epochs earlier than the other models, reducing computation by 11–28%. Furthermore, Section 3.3 shows that reducing the batch size allows the 2D-Transformer to converge to lower values of the loss function within fewer epochs as PGW-Lite.

Figure 5 shows the development of the RMSE for different variables as the model continues to learn for the PGW-Lite model trained on a global minibatch size of 720. For the first 230 epochs, the learning rate is kept constant at $5 \cdot 10^{-4}$. As paralleled in the validation loss plots in Figure 4, the model learns relatively slowly for the first 100 epochs except for the initial drop. After that, particularly notable in the velocity and geopotential fields, the model is able to learn more quickly before reaching a second plateau. Though it is not shown here, we observe similar patterns for all ablated models.



**Table 2.** Best validation loss and total epochs for the different models

| Model | Best Validation loss | Epoch # |
|-------|---------------------|---------|
| PGW-Lite | 0.143 | 360 |
| 2D-Transformer | 0.148 | 263 |
| Relative Bias | 0.143 | 300 |
| Positional embedding | 0.152 | 358 |
| Three-depth | 0.143 | 346 |

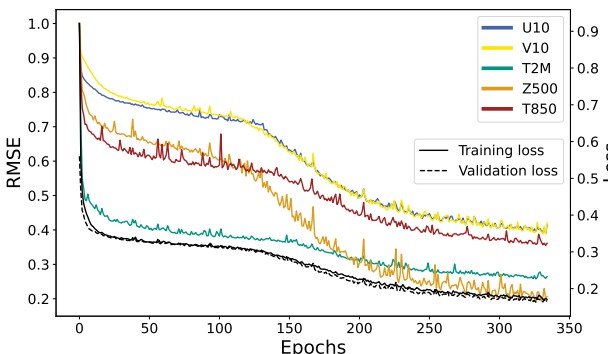

**Figure 5.** RMSE as a function of epochs for different prognostic variables for the PGW-Lite, as evaluated on the validation dataset. All values are normalized by their maximum RMSE value.

## 3.3 Minibatch size effects

Figure 6 shows the validation loss of two models, PanguLite and 2D attention, with a global minibatch size of 16, 32, 64, 480, and 720. More experiments could not be conducted due to computational constraints. The details of the local and global minibatch size, as well as the average total GPU-hours/epoch and wall time per epoch, are shown in Table 3. All models were initialized with the same random seed, so the order of the training samples loaded from the data loader are identical. Both models train nearly identically for a minibatch size of 16, but diverge for all other minibatch sizes. For the 2D-Transformer model, similar final loss values were reached for minibatch sizes of 16, 64, and 480. The 2D-Transformer model appears to have instabilities in the beginning of training for a batch size of 32, leading it to converge at a worse final loss value. For the 2D model, it appears that using a small global minibatch size reduces the time to convergence and improves the final model performance. In contrast, the PGW-Lite model failed to converge for minibatch sizes of 32, 64, and 480. In these cases, the models were allowed to train until the learning rate dropped to $3 \cdot 10^{-5}$.

The global minibatch size greatly affects the speed of convergence: by training with a smaller global minibatch size, we reach lower values of the loss function much more quickly; this also comes with a reduction in overall compute power, but comes at the cost of longer wall times.



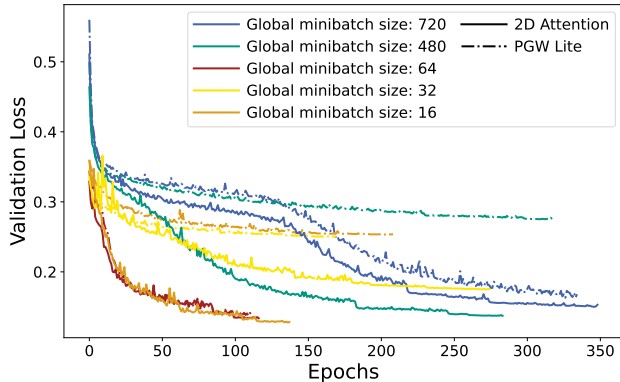

**Figure 6.** Validation loss of 2D-Attention and PGW-Lite as a function of epochs for global minibatch sizes of 16, 32, 64, 480, and 720.

**Table 3.** Compute requirements for PG Lite and 2D based on minibatch size.

| Global minibatch size | Local Batch size | | Average total GPU-hours/epoch | | Wall time/epoch [min] | |
|---|---|---|---|---|---|---|
| | PG Lite | 2D | PG Lite | 2D | PG Lite | 2D |
| 720 | 6 | 12 | 11.3 | 10.6 | 5.6 | 10.6 |
| 480 | 6 | 12 | 9.2 | 8.0 | 6.9 | 12.0 |
| 64 | 1 | 1 | 10.6 | 8.6 | 11.3 | 8.1 |
| 32 | 1 | 1 | 10.9 | 7.8 | 20.5 | 14.7 |
| 16 | 1 | 1 | 8.5 | 7.7 | 31.7 | 28.8 |

## 3.4 Variable-specific weighted cosine loss scheduling

Figure 7 shows the RMSE of variables U10, V10, T850, and T2M over the course of the cosine loss scheduling. The final L1 loss value reached was 0.131, lower than all other models trained in this paper. The large fluctuations in the training loss reflect the cosine scheduling of the different variables. The RMSE of the different key variables are shown along side the training and validation loss in Figure 7.

Despite the large fluctuations in the training loss, the validation loss continues to decrease. There are some fluctuations in the RMSE variables, reflecting the effects of the weight scheduling: between epochs 125 and 175, the weights for the velocity fields are small. In this period, we notice that the validation RMSE for the velocity variables increase slightly as the temperature, weighted highly, decreases.





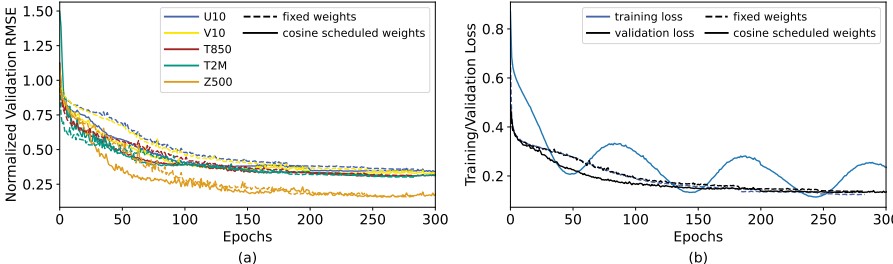

**Figure 7.** Error metrics for variable-specific weighted cosine loss scheduling applied to the 2D-Transformer model. PGW-Lite was maintained as the reference model and compared to the variable-specific weighted cosine loss scheduling model. (a): RMSE values of the 10m-U- and V-velocity, as evaluated on the validation dataset, as a function of the epochs. (b): training and validation losses for the two models.

## 4 Discussion

### 4.1 Ablation Study

The major finding of this ablation study is that neither of the two novel ideas introduced by PGW are crucial to its success. This underscores the idea that the success of PGW is due to the Transformer backbone and SWIN mechanism, rather than

the particular bias term introduced or the 3D-Transformer architecture. This is noteworthy because these two architectural components are particularly expensive with respect to training time and memory requirement. In fact, the PGW-Lite model does not perform better than the other ablated, smaller models. The largest and deepest model, Three-depth, was hypothesized to perform much better, since the individual windows in SWIN would perform attention on farther ranges in the deepest layers, meaning that far-reaching weather patterns can also be learned. Surprisingly, the model appears to be difficult to train and have

convergence issues; the final performance of Three-depth was not better than PGW-Lite. The results of this study show that the massive ESB term can be replaced by a simple learned positional embedding, reducing the size of the term that learns "earth-specific" locality by 20.4 million parameters, even in the case of PGW-Lite; in the case of the full PGW model, this term contains 40.4 million of the total 64.2 million parameters.

Table 2 and Figure 4 show that the 2D-attention mechanism is a suitable replacement for the 3D-Transformer. They show

that the 2D-Transformer learns the representation faster than any of the 3D models, reaching a comparable validation loss and RMSE value in fewer epochs. Intuitively, we expect that the 3D-Transformer introduced in Bi et al. (2023a) should perform better than the 2D-Transformer. However, evaluating these results shows that the 2D-Transformer is sufficient. The interpretation is: in the architecture, after patch embedding, a representation of all variables in each patch is learned and stored in a latent vector of dimension $C$. With a 3D-Transformer architecture, each latent vector contains the information of the five pressure

variables (Z, Q, T, U, V) and attention is performed between neighboring patches in all dimensions. In a 2D-Transformer architecture, the latent vector contains the five pressure variables at all pressure levels, and attention is only performed over the neighboring patches in the latitude and longitude dimensions. It is possible that mixing the information across all spatial layers is more efficient, as it allows the model to learn the relationship between more than two pressure levels at a time. This has





many benefits—by using 2D-attention, the individual attention blocks require much less memory. Specifically, the attention mechanism requires an $O(n^2)$ memory requirement with respect to the size of the sequence (Vaswani et al., 2017). Reducing the attention blocks to perform attention over patches from (2, 6, 12) to (1, 6, 12) in the 2D-Transformer has allowed us to double the local minibatch size per GPU, reducing the total computational requirements.

Due to the infinitely large hyperparameter space and apparent sensitivity of the model to these parameters, it is unclear whether the presence of the 3D-Transformer or ESB is truly detrimental to the convergence and success of the model. It is entirely possible that, given the correct hyperparameter configuration, the original PGW model could greatly outperform these cheaper, ablated models. However, due to the cost, energy requirements, and associated $CO_2$ emissions of the compute requirements of the original model, we argue that it is important to develop easily trainable and robust models; we strive for a model with little hyperparameter tuning, reducing the compute requirements of training. Figure 4 and 6 show that all models follow two phases of training: after a certain point, the model learns a suitable embedded representation and can proceed to learn more quickly. In this work, we explore different mechanisms to reduce the number of epochs it takes these models to reach this trigger point; this reduces the computational burden and makes these data-driven models accessible to researchers with smaller compute budgets.

## 4.2 Batch size

The difficulty and unpredictability of training Transformers is shown in Figure 6, where the PGW-Lite and 2D-Attention models were trained for different global minibatch sizes of 16, 32, 64, 480, and 720. All models were initialized with the same random seed. With the exception of minibatch size 16, the 2D-Attention model trained within fewer epochs than the equivalent PGW-Lite model. While the 2D-Attention minibatch 32 model also showed instability and difficulty training, PGW-Lite with minibatches of 32, 64, and 480 failed to converge and leave the "phase 1" stage of training; Despite the learning rate dropping to small values of $3 \cdot 10^{-5}$, the models did not improve and training was cancelled. The results indicate that the higher model complexity with PGW also increase difficulty in convergence. Given that many researchers face limited compute budgets, this underscores the benefit for having a model robust to training and hyperparameters.

Table 3 shows that the 2D models consistently require fewer total GPU-hours and wall time to train than the PGW-Lite models, despite having more model parameters. This highlights the computational cost of the attention mechanism, as the PGW-Lite model performs attention across windows that are $2\times$ larger than the 2D-Attention model. For both models, doubling the GPUs results in a speedup of between 80–95%, revealing the increased synchronization time required with more parallelization. When comparing the run times for global minibatch sizes of 480 and 720, the 2D-Attention model is able to fit $2\times$ the number of samples per GPU. This comes at the expense of wall time, where we see the 2D models requiring between 1.5–1.7$\times$ more wall time to complete a given epoch, but similar overall computational demands.

The relationship between Average Total GPU-hours/epoch and batch size exhibited by the 2D-Attention model show that using smaller minibatch sizes will reduce the computational time per epoch, at the cost of wall time. Combined with the speedup in convergence behavior, i.e., smaller global batch size leads to fewer overall epochs required, the results indicate the benefits





of using both smaller local and global batch sizes when training weather forecasting models. Our limited results emphasize the need for more systematic studies on the relationship between local and global batch size, wall time, and convergence.

### 4.3 Variable-specific weighted cosine loss scheduling

This study highlights the sensitivity of Transformer-based medium-range weather forecasting models to hyperparameters such as batch size and learning rate. With this in mind, we wanted to develop some explainable heuristics that could be applied to the training scheduling. The advantage of training an autoregressive weather forecasting model is that the input and output fields are the same, and the fields describe physical dynamics. We look at the RMSE over epochs in Figure 5 to understand how the model learns. We notice a few trends: as expected, the U- and V-velocity fields learn at the same rates. The temperature fields

level off more quickly, whereas the geopotential field continues to improve as training goes on, even when the loss function appears to have stagnated.

The design behind the variable-specific weighted cosine scheduling is because it was observed that the velocity field, compared to fields such as geopotential or temperature, take longer to learn. This is attributed to the sharp and localized gradients that occur in these fields, making them harder to learn and generalize. However, in atmospheric models, resolving the velocity

field is crucial, since velocity is responsible for major atmospheric processes such as advection. This causes the wind velocity to be a major driver for other atmospheric variables. As such, we decided to schedule the weights in such a way that the velocity fields are encouraged to improve quickly in the beginning, at the cost of learning other fields such as the temperature in the beginning.

Figure 7 shows that even as the training loss fluctuates due to a variable-specific weighted scheduling, the validation RMSE

values of different variables will generally continue improving, with only minor fluctuations. Particularly in the beginning of training, the implementation of the variable-specific weighted cosine loss scheduling allows the model to converge with approximately 30% fewer epochs, with all other hyperparameters and training regimens kept constant.

### 5 Conclusions

This work presents an ablation study of PGW to determine the important architectural components that lead to its success.

It finds that the ESB can be replaced with a simple relative bias, reducing the overall number of model parameters by 46%. We also show that replacing the 3D-Transformer with a 2D-Transformer, even though it has 30% more parameters, allows for the model to fit more samples on a single GPU—in our case, 2× that of the former. Furthermore, the 2D models consistently train faster or just as fast as the the three-dimensional models and reach better L1 and RMSE values upon convergence. An initial study on the batch size effects of these models shows that the 2D-Transformer is much more robust to training than

PGW-Lite, which often failed to converge. We also show that training either model with a global minibatch size of 16 with a local batch size of 1 can allow the model to converge within 28% of the total epochs compared to a minibatch size of 720, while maintaining similar total GPU-hours/epoch and increasing the wall time sub-linearly. This points to further work in understanding the relationship between local and global batch size, total GPU-hours required, and wall time. We also show



that implementing a simple weighted loss function schedule over time can push the model to converge with nearly 30% fewer epochs, effectively reducing the computational cost to reach a given performance at no change to the model architecture or training schedule.

*Code availability.* The current code version is available from the project website: https://doi.org/10.5281/zenodo.11601007.

*Author contributions.* DT and CD conceptualized the project; JQ curated the data; DT developed the software and conducted experiments and formal analysis; GAH and CD acquired funding; GAH, CD and AS supervised the project; All authors contributed to the writing and editing of the paper.

*Competing interests.* The authors declare that they have no conflict of interest.

*Acknowledgements.* This work was supported by the KIT Center MathSEE and the KIT Graduate School for Computational and Data Science under the FAST-DREAM Bridge PhD grant and by the German Federal Ministry of Education and Research under the 01LK2313A - SMARTWEATHER21-SCC-2 grant. The authors gratefully acknowledge the computing time made available to them on the high-performance computer HoreKa at the NHR Center KIT via the SmartWeather21-p0021348 NHR large project. This center is jointly supported by the Federal Ministry of Education and Research and the state governments participating in the NHR (www.nhr-verein.de/unsere-partner).

This work was performed on the HoreKa supercomputer funded by the Ministry of Science, Research and the Arts Baden-Württemberg and by the Federal Ministry of Education and Research.



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
