# Peer review of "Architectural Insights and Training Methodology Optimization of Pangu-Weather"

_EGUsphere, 2024_

## Author Comment (AC2)

**Response to Reviewer**

August 8, 2024

**Detailed Comments**
*Responses are marked in blue.*

- p. 2: new training procedure 30% faster - compared to 2D or original 3D?

  We apologize for the lack of clarification on that part. The improvement in training speed is compared to the 2D model, run with the exact same training configuration (same learning rate, same initial seed, same local and global batch size). We have clarified this in the text with the following phrasing:

  ...we propose a new training procedure that increases the speed of convergence for the 2D-Transformer model (compared to the original 2D model) by 30% without adjusting any other hyperparameters, increasing the possible performance of the model given a fixed compute budget."

- p. 3: what was the number of compute nodes? were local SSDs used in some form? if not, is mentioning them relevant to comparable studies (I believe such hybrid setups would be very peculiar to use)?

  The number of nodes was not fixed but is given through the number of GPUs utilized, i.e. the global batch size. We apologize for not stating this more clearly. Each compute node is equipped with four A100 GPUs. As stated in section 2.4, *"The validation case was trained with a local batch size of two on 120 A100-40 NVIDIA GPUs. Unless otherwise specified, all ablated models were trained with a local batch size of six on 120 A100-40 NVIDIA GPUs"*.

  Runs always utilized all available GPUs on a node, splitting batches across them as evenly as possible. Hence, the number of nodes utilized for ablated models can be derived from the global batch size, divided by the local batch size (for ablation models, = 6) , divided by four (number of GPUs per node). The term "Ablated models" here comprises the models presented in Section 3.2 (Ablation study), but not the mini-batch experiments in Section 3.3 (Minibatch size effects). The local batch size of the minibatch study can be found in Table 3.

  The local SSDs were not used in this study. We agree with the reviewer that it would be interesting to investigate using them for optimized data I/O (Data staging from Lustre to local SSDs, binding all local SSDs using Beyond to have true shuffling between epochs, and then loading from there); this could also yield significant speed-up. However, such an implementation requires a substantial amount of platform-specific adaptation and would not

be translatable to other systems, hence reducing comparability for other users. We have thus removed the statement about on the local SSDs from the description of the compute environment.

- p. 5, fig. 1b: These plots do not show perfect sin/cos functions, they are skewed; reading the explanation in 2.6, I don't understand why. I believe this comes from tweaking them to match the original weight sums, but then I'm missing an explanation for this particular tweak. Also, they are not in phase as explained (l. 134); e.g., the maxima of U/V are slightly shifted (epoch 200 in (a), epoch 185 in (b)).

We thank the reviewer for their observation and apologize for the confusion. The initial weights are distributed according to normal cosine functions. We also design the cosine distributions such that the velocity fields in the surface fields peak at the same time as in the pressure levels. However, at every epoch, we need to normalize the sum of the weights to match those of the original weight sums.

The reason that the peaks are slightly offset is due to this weight normalization. Taking the U/V in plot (b) for instance: according to the regular cosine scheduling, the peaks should be at epochs 0, 100, and 200. However, since the weight of MSP at these points has a higher proportion of the overall weight, the relative weight of U/V is decreased. At epoch 85, the weights of MSP and T2M are very low, meaning that the U/V scales to a greater value.

We have included a more explicit formulation of the weight distribution in the manuscript as follows:

"Given an individual variable $i$ at epoch $e$, for a period of $n_{\text{epochs}}$ (in this case, $n_{\text{epoch}} = 100$). For the pressure variables Z, Q, T, U, V, $i = 1, 3, 2, 0, 0$, respectively. For the surface variables MSLP, U10, V10, T2M, $i = 1, 0, 0, 2$, respectively.

$$w(i, e) = \cos\left(\frac{2\pi}{n_{\text{epoch}}}\left(e - \frac{i}{2}\right)\right) + 1.1 \tag{1}$$

At each epoch, the pressure variables are normalized to sum up to the original weights of $3.00 + 0.60 + 1.50 + 0.77 + 0.54$.

$$W(i, e) = w(i, e) \cdot \frac{3.00 + 0.60 + 1.50 + 0.77 + 0.54}{\sum_i w(i, e)} \tag{2}$$

A similar procedure is applied to the surface variables, where the numerator is replaced with $1.5 + 0.77 + 0.66 + 3.0$.

The ordering of the weights was designed such that the weights of the velocity and temperature variables (U, U10; V, V10; T, T2M) would peak at similar epochs."

- p. 8, l. 172: If I read figure 6 correctly, PGW-Lite failed to converge for sizes of 16, 32, and 480. For 64, it converged (hard to read the figure here but I believe there's a dashed red line just behind the solid red/orange line). This is in contradiction to what is written in the text.

Thank you for your correction; we apologize for this error and we have updated this in the text:

"In contrast, the PGW-Lite model converges at sub-optimal loss values for minibatch sizes of 16, 32, and 480."

We have also split up the figure into two subfigures and changed the color scheme consistently across the manuscript to make the plots more legible.

[Figure]

**Figure 6.** Validation loss of 2D-Attention and PGW-Lite as a function of epochs for global minibatch sizes of 16, 32, 64, 480, and 720.

- p. 8: The point of the minibatch study appears to be to 1.) analyze how minibatch sizes affect attainable loss/convergence and 2.) make a direct comparison on this between the 2D and 3D transformer approaches. For me as reader it would have been good to point this out here (it only became clear when reading discussion and conclusion) because it affects how one reads the text and plot.

  Thank you for the observation—we have introduced a new subsection (Section 2.6) in the methodology that motivates and explains the minibatch study:

  "A study directly comparing the PGW-Lite model, which features a 3D-Attention mechanism, and the 2D-Attention model was performed. Each of the two models was trained from scratch five times, varying the global batch size from 16, 32, 64, 480, 720. More experiments could not be conducted due to computational constraints. The details of the local and global minibatch size, as well as the average total GPU-hours/epoch and wall time per epoch, are shown in Table 3. All models were initialized with the same random seed, so the order of the training samples loaded from the data loader are identical."

- p. 8: Would different random seeds have a significant effect on convergence/attainable loss?

  We thank the reviewer for this question. Given the irregularity in the final loss values for the PGW-Lite model, we hypothesize that the use of different random seeds can have a significant effect on the attainable loss. Due to computational constraints, we did not perform multiple repetitions to study the effect of different seeds on the final loss values of these models. To address this, we have added the following explanation to Section 3.3:

  "In contrast, the PGW-Lite models converge at sub-optimal loss values for minibatch sizes of 16, 32, and 480. In these cases, the models were allowed to train until the learning rate dropped

to $3 \cdot 10^{-5}$. The irregularity of this behavior can be attributed to the model's sensitivity to the initial random seed."

- in general, while zooming helps, the colour scheme (use of yellow) and size of figures 6 and 7 makes them hard to read, particularly given that many lines relevant to discussion overlap.

Thank you for your feedback. We have increased the size of all figures, particularly the ones with subfigures. We have also changed the color scheme to remove the yellow. As stated above, we have split up Figure 6 into two subfigures for clarity. We have also reduced the y-axis range on Figure 7 to show the plot more clearly.

[Figure]

**Figure 6.** Validation loss as a function of epochs for global minibatch sizes of 16, 32, 64, 480, and 720 of (a): 2D-Attention and (b): PGW-Lite.

[Figure]

**Figure 7.** Error metrics for variable-specific weighted cosine loss scheduling applied to the 2D-Attention. The 2D-Attention model with PyTorch's ReduceLROnPlateau scheduler was maintained as the reference model and compared to the variable-specific weighted cosine loss scheduling model. (a): RMSE values of the 10m-U- and V-velocity, as evaluated on the validation dataset, as a function of the epochs. (b): training and validation losses for the two models.

---

## Author Comment (AC3)

**Response to Reviewer**

August 8, 2024

*Responses are marked in blue.*

**General comments**

- Figure 3 is an important chart supporting the validity of this research. However, it does not include specific humidity, Z500, or V10, as mentioned earlier in this paper. Including these variables could strengthen the argument for the effectiveness of the 2D-Transformer. Due to the 6-hour subsample, readers ultimately do not know if the 2D-Transformer has improved the forecast accuracy of the original Pangu-Weather model. Including such comparisons could significantly increase the citation rate of this paper. There are still certain changes and clarifications that the authors should address prior to publication. For these reasons, I believe that the manuscript can be accepted for publication. Below, I have some specific comments to the authors.

  Thank you for your insight. The RMSE values from the official Pangu-Weather are obtained from their summary performance information in their Github repository (Bi et al., 2023), where detailed performance metrics of Z500, U10, T2M, and T850 are reported. V10 is missing from this table. We have included the evaluation of Z500 in Figure 3, indicated by the yellow line. Our version of Pangu did not outperform IFS for Z500. We have modified the text accordingly:

  "We observe that our PGW model performs better than IFS for three variables (U10, T2M, T850) over the five-day forecast but performs worse for the Z500 variable. There is still a notable difference between the performance of our model and the published PGW model. We attribute that to training the model on only a 6 h-subsample of the total data trained by the original authors, as well as slight differences in training procedure such as batch size."

[Figure]

**Specific comments**

- Line #2 - #5, the sentence is too long and difficult to read. It can be revised to "The Transformer-based PGW introduced novel architectural components, including the three-dimensional attention mechanism (3D-Transformer) in the Transformer blocks. Additionally, it features an Earth-specific positional bias term that accounts for weather states being related to the absolute position on Earth."

  Thank you for the suggestion. We have revised the sentence "The Transformer-based PGW introduced novel architectural components including the three-dimensional attention mechanism (3D-Transformer) in the Transformer blocks and an Earth-specific positional bias term..." to "The Transformer-based PGW introduced novel architectural components including the three-dimensional attention mechanism (3D-Transformer) in the Transformer blocks. Additionally, it features an Earth-specific positional bias term..."

- Line #24, "the authors also admit" could be replaced with more specific wording, such as "previous studies have shown". The same issue appears in Line #91, where the architecture described "by the authors" could be replaced with "in this study." This sentence reads as if the ablation study is original to this paper and not derived from the model itself. If this is the case, some references could be cited here as evidence to support the experiment design.

  Thank you for your comment. "The authors", in this case, refers to Bi et al., who published the original Pangu-Weather model. Following your suggestion, we have changed line 24 to "The authors of PGW admit that the models have not arrived at full convergence (Bi et al., 2023a)".

  We have also changed Line 96 to specifically reference the authors of Bi et al. as follows:

"...an ablation study was performed based on the PGW-Lite architecture described by Bi et al. (2023) rather than the full PGW model."

- Line #27, the published model cannot be run, what is the reason? Is it also caused by modularized manner issue? Does it conflict with reproduction introduced in section 2.3?

  Thank you for this question. The reason why the published model code cannot be run is that the original authors Bi et al. (2023) did not publish their model training code, but only an inference code that is based on the already trained model weights. The pseudocode that is published by the authors outlines the general architecture of the model, but is written in a way that requires heavy modification and re-implementation before the code can be run. Furthermore, all other publicly available re-implementations of Pangu-Weather are either less complete (in terms of implementation or documentation) than ours, and would have required significant modification to perform our targeted ablation study.

  We have modified the body of the text as follows: "The pseudocode outlines the architecture, but is not complete Python code that can be run without major modification."

- Figure 1, 4, 5 and 7 could be appropriately enlarged. Some images are difficult to discern even when enlarged. For Figure 4 (a), the authors can separate the lines by adjusting the y-coordinates.

  We apologize for this and thank you for the suggestions. Based on your feedback, we have changed the y-axis to increase the readibility of Figure 4.

[Figure]

**Figure 4**: Training curves of ablation study for PGW-Lite, 3-depth model, relative bias, positional embedding, and 2D-Attention models. (a) Training and validation loss values as a function of the epochs. (b): RMSE for the 10m-U-velocity as a function of the epochs, as evaluted for the validation samples.

We have increased the size of Figures 1, 4, 5, 6, and 7.

We have also split up Figure 6 into two subfigures to facilitate parsing the information in the plots more easily:

[Figure]

[Figure]

**Figure 6.** Validation loss of 2D-Attention and PGW-Lite as a function of epochs for global minibatch sizes of 16, 32, 64, 480, and 720.

- Figure 1, the cosine functions for each variable with weights from Bi et al. (2023) could be listed to explain the normalized process at each epoch. In Figure 1(b), it would be helpful to present the equation for the sloped MSP graph to facilitate understanding.

We thank the reviewer for their observation and apologize for the confusion. The initial weights are distributed according to normal cosine functions. We also design the cosine distributions such that the velocity fields in the surface fields peak at the same time as in the pressure levels. However, at every epoch, we need to normalize the sum of the weights to match those of the original weight sums.

The reason that the peaks are offset is due to this weight normalization. Taking the U/V in plot (b) for instance: according to the regular cosine scheduling, the peaks should be at epochs 0, 100, and 200. However, since the weight of MSP at these points has a higher proportion of the overall weight, the relative weight of U/V is decreased. At epoch 85, the weights of MSP and T2M are small, meaning that the U/V scales to a greater value.

We have included a more explicit formulation of the weight distribution in the manuscript as follows:

"Given an individual variable $i$ at epoch $e$, for a period of $n_{\mathrm{epochs}}$ (in this case, $n_{\mathrm{epoch}} = 100$). For the pressure variables Z, Q, T, U, V, $i = 1, 3, 2, 0, 0$, respectively. For the surface variables MSLP, U10, V10, T2M, $i = 1, 0, 0, 2$, respectively.

$$w(i, e) = \cos\left(\frac{2\pi}{n_{\mathrm{epoch}}}\left(e - \frac{i}{2}\right)\right) + 1.1 \tag{1}$$

At each epoch, the pressure variables are normalized to sum up to the original weights of $3.00 + 0.60 + 1.50 + 0.77 + 0.54$.

$$W(i, e) = w(i, e) \cdot \frac{3.00 + 0.60 + 1.50 + 0.77 + 0.54}{\sum_i w(i, e)} \tag{2}$$

A similar procedure is applied to the surface variables, where the numerator is replaced with

$1.5 + 0.77 + 0.66 + 3.0$. The ordering of the weights was designed such that the weights of the velocity and temperature variables (U, U10; V, V10; T, T2M) would peak at similar epochs."

- Figure 6: Could the author explain the reason for the failure to converge based on PGW-Lite structure with minibatch sizes of 32, 64, and 480? PGW-Lite with minibatch size 720 eventually converged. Could the authors explain this unexpected result? Part of the reason is explained in Line #229. It is not necessary to strictly separate the results and discussion sections. Explaining part of the findings in the result section can enhance the content.

  We apologize for the lack of detail in our original submission regarding these results and their interpretation. We gladly shed some more light on that aspect: Given the irregularity in the final loss values for the PGW-Lite model, we hypothesize that the use of different random seeds can have a significant effect on the attainable loss. Due to computational constraints, we did not perform multiple repetitions to study the effect of different seeds on the final loss values of these models. To address this, we have added the following explanation to Section 3.3:

  "In contrast, the PGW-Lite models converge at sub-optimal loss values for minibatch sizes of 16, 32, and 480. In these cases, the models were allowed to train until the learning rate dropped to $3 \cdot 10^{-5}$. The irregularity of this behavior can be attributed to the model's sensitivity to the initial random seed."

- Line #255, the sentence could be updated to "since wind vectors, acting as pressure gradients, can drive certain atmospheric processes, such as advection terms in atmospheric variables."

  We thank the reviewer for their suggestion in improving our phrasing. We have implemented the suggestion in the body of the text:

  "This causes the wind velocity to be a major driver for other atmospheric variables since wind vectors, acting as pressure gradients, can drive certain atmospheric processes, such as advection terms in atmospheric variables"

**Other suggestions**

Following are suggestions and do not affect the validity of the argument in this paper.

- Line #98: the models could be compared in more details in Table 1, like the hidden dimension, etc.

  We thank the reviewer for their suggestion. To enhance the clarity, we have followed the suggestion and added the hidden dimension into Table 1.

Table 1: Overview of models included in ablation study. All models are based on PGW-Lite.

| Model | Number of parameters (millions) | Hidden dimension |
|---|---|---|
| Pangu-Weather-Lite (absolute bias) | 44.6 | 192 |
| Relative bias | 24.3 | 192 |
| Positional embedding | 24.3 | 192 |
| 2D-Attention | 57.2 | 288 |
| Three-depth | 108.9 | 192 |

**Technical corrections**

- Figure 5 y axis could be updated into "normalized RMSE"

  Thank you for your detailed observation. We have modified the figure accordingly.

[Figure]

**Figure 5.** RMSE as a function of epochs for different prognostic variables for the PGW-Lite, as evaluated on the validation dataset. All values are normalized by their maximum RMSE value.

---

## Author Comment (AC4)

**Response to Reviewer**

August 8, 2024

*Responses are marked in blue.*

**Comments**

- The model names used in the tables and in the text are not consistent. Table 1 and Table 2 might be merged with unique model names.

  We apologize for our oversight. We have now consolidated the names of the models within Tables 1 and 2.

**Table 1**: Overview of models included in ablation study. All models were based on PGW-Lite.

| Model | Number of parameters (millions) | Hidden dimension |
|---|---|---|
| Pangu-Weather-Lite (absolute bias) | 44.6 | 192 |
| Relative bias | 24.3 | 192 |
| Positional embedding | 24.3 | 192 |
| 2D-Attention | 57.2 | 288 |
| Three-depth | 108.9 | 192 |

**Table 2:** Best validation loss and total epochs for the different models

| Model | Best Validation loss | Epoch # |
|---|---|---|
| Pangu-Weather-Lite (absolute bias) | 0.143 | 360 |
| Relative Bias | 0.143 | 300 |
| Positional Embedding | 0.152 | 358 |
| 2D-Attention | 0.148 | 263 |
| Three-depth | 0.143 | 346 |

- Visualisation of the modification in model architecture: I find the modularised code in the repository very helpful. Could the code be included as pseudo code in the paper giving a comparative overview of the architectural details of the different models? This would be helpful in connection with the graphics in the original Pangu Publication (Fig. 2).

  Thank you for your comment. We have provided a visualization of the different architectures

in the following figure. For simplification and clarity's sake, these diagrams neither include the up and downsampling layers, nor the Sliding-Window Transformer (SWIN) architecture. This is also why the Three-depth model is not depicted here, as its architecture is largely the same as Pangu-Weather Lite, albeit with an additional up/downsampling layer.

[Figure]

(a) Pangu-Weather

(b) Relative bias

(c) Learned positional embedding

(d) 2D-Attention

**Figure:** Architecture of Pangu-Weather model and variations considered in the ablation study. The sliding window mechanism (SWIN) is not depicted in the figure for simplicity. The up- and downsampling layers are also excluded from the figure.

- Parameter numbers in Table 1: The numbers of parameters for the 2d attention model s larger than for the 3d attention (PanguLite) in Table 1. This is counter intuitive. Is it due to the fact that the hidden dimension C was enlarged? What was the reasoning behind that choice? Could it be chosen such that the overall parameter size would match that of PanguLite again? Is this dimension C the same for PanguLite and Pangu? Could the authors extrapolate the parameter numbers in Table 1 for the original Pangu model with the original batch size?

  Thank you for your insight. This is explained in the end of Section 2.5 with the following text:

  "To compensate for the reduction in parameters associated with a 2D-Transformer model, the hidden dimension was increased from C = 192 to C = 288, increasing the dimension of the hidden layer to 48 per attention head as opposed to the original 32. Note that the latent space in the 2D-Transformer now encodes 5 variables × 13 pressure levels instead of just 5 variables × 1 pressure level as in the 3D case."

  To enhance the clarity, we have followed your suggestion and added the hidden dimension into Table 1.

Table 1: Overview of models included in ablation study. All models were based on PGW-Lite.

| Model | Number of parameters (millions) | Hidden dimension |
|---|---|---|
| Pangu-Weather-Lite (absolute bias) | 44.6 | 192 |
| Relative bias | 24.3 | 192 |
| Positional embedding | 24.3 | 192 |
| 2D-Attention | 57.2 | 288 |
| Three-depth | 108.9 | 192 |

- Parameter numbers in Table 3 (relating to Remark 3): In paragraph 2.5 the authors state that reducing the model size allows for larger local batches. Hence, PanguLite should have larger local batches than the 2d version. In Table 3 it is the other way round. Could the authors please clarify this?

  We apologize for the confusion. While it is true that increasing the model size will allow for more data samples to fit on a single GPU, the main memory cost in training Transformer models comes from the attention block. When the models are performing the forward and backward passes, many intermediate states need to be retained in memory on the GPU—this explains why even though the model weights have a size of about 500 MB, and input data has a size of about 250 MB, 20 GB of memory is required on the GPU during one forward and backward pass. As the 2D-Attention mechanism essentially halves the memory requirement of attention (since attention is computed over $1 \times 6 \times 12$ windows instead of $2 \times 6 \times 12$), 2D-Attention requires less memory on the GPU.

  This is explained in section 4.1 as follows:

  "by using 2D-attention, the individual attention blocks require much less memory. Specifically, the attention mechanism requires an $\mathcal{O}(n^2)$ memory requirement with respect to the size of the sequence (Vaswani et al., 2017). Reducing the attention blocks to perform attention over

patches from (2, 6, 12) to (1, 6, 12) in the 2D-Transformer has allowed us to double the local minibatch size per GPU, reducing the total computational requirements."

- Fig. 1: As the curves for U and V are indistinguishable, one colour for both curves would render the figure clearer.

  Thank you for your attention to detail. We have combined the color of U and V (in subfigure (a)) and the color of U10 and V10 (in subfigure (b)).

- Furthermore, plot a) and b) should display temperature and wind in the same colour.

  We have chosen to use different colors for T and T2M; and U and U10; because these colors are consistent with earlier figures from the rest of the manuscript, i.e., Figure 1 and 3.

[Figure]

(a)  (b)

**Figure 1**. The weights for variable-specific cosine loss scheduling. The period is 100 epochs. The U- and V-velocity fields for both the pressure-level variables and surface variables are superimposed on one another, since they always receive the same values. The weights are normalized to sum up to the sum of the original weights presented in Bi et al. (2023a). (a) Pressure-level variables. (b) Surface variables.

[Figure]

**Figure 3.** RMSE as a function of forecast lead time of key prognostic variables (T2M, U10, T850) in our model compared to IFS and Pangu-Weather on testing data from 2020–2021.

- Figure 7: What does it mean that the reference model is PanguLite? The text implies that the two curves show both the 2d model with different training losses.

  We apologize for the mistake. We have updated the figure caption to the following:

  "Error metrics for variable-specific weighted cosine loss scheduling applied to the 2D-Attention model. The 2D-Attention model with PyTorch's ReduceLROnPlateau scheduler was maintained as the reference model and compared to the variable-specific weighted cosine loss scheduling model. (a): RMSE values of the 10m-U- and V-velocity, as evaluated on the validation dataset, as a function of the epochs. (b): training and validation losses for the two models."